# Interfacial dynamics mediate surface binding events on supramolecular nanostructures

Ty Christoff-Tempesta [1,6,8], Yukio Cho [1,7,8], Samuel J. Kaser[2], Linnaea D. Uliassi[1], Xiaobing Zuo [3], Shayna L. Hilburg [4], Lilo D. Pozzo [4] & Julia H. Ortony[1,5] ✉

The dynamic behavior of biological materials is central to their functionality, suggesting that interfacial dynamics could also mediate the activity of chemical events at the surfaces of synthetic materials. Here, we investigate the influence of surface flexibility and hydration on heavy metal remediation by nanostructures self-assembled from small molecules that are decorated with surface-bound chelators in water. We find that incorporating short oligo(ethylene glycol) spacers between the surface and interior domain of self-assembled nanostructures can drastically increase the conformational mobility of surface-bound lead-chelating moieties and promote interaction with surrounding water. In turn, we find the binding affinities of chelators tethered to the most flexible surfaces are more than ten times greater than the least flexible surfaces. Accordingly, nanostructures composed of amphiphiles that give rise to the most dynamic surfaces are capable of remediating thousands of liters of 50 ppb $Pb^{2+}$-contaminated water with single grams of material. These findings establish interfacial dynamics as a critical design parameter for functional self-assembled nanostructures.

The dynamics of biological soft matter systems play a critical role in enabling molecular recognition and binding[1,2]. For example, the dynamics of protein relaxation and fluctuation events control their ability to bind ligands[3,4]. These findings have replaced the static lock-and-key models for protein binding events with more complete representations, such as the induced fit[4] and conformational selection models[5]. As a result, the influence of molecular flexibility—a critical design parameter for controlling soft matter dynamics − on the binding affinities of biological systems is now considered a 'fundamental determinant of intermolecular interaction strength'[1]. Despite progress in the past few decades towards uncovering dynamics-function relationships in biological systems[6], designing for molecular flexibility is largely overlooked on the surfaces of synthetic soft matter systems.

The dynamics of water at and around soft matter interfaces are also intricately coupled to the chemical identity of the interface and can in turn mediate surface chemical events[7–9]. For example, interfacial water plays an active role in protein function, including mediating protein binding and folding[10,11]. Investigations into the role of water at and around biomacromolecules have uncovered three 'types' of water based on its rate of translational motion: bulk water, hydration water, and structural water. Whereas bulk water far from macromolecular interfaces behaves independently of solute influences, hydration and structural water experience suppressed diffusion from their

[1]Department of Materials Science and Engineering, Massachusetts Institute of Technology, Cambridge, MA, USA. [2]Department of Chemistry, Massachusetts Institute of Technology, Cambridge, MA, USA. [3]X-ray Science Division, Advanced Photon Source, Argonne National Laboratory, Lemont, IL, USA. [4]Department of Chemical Engineering, University of Washington, Seattle, WA, USA. [5]Department of Chemistry and Biochemistry, University of California San Diego, La Jolla, CA, USA. [6]Present address: Department of Chemical and Biomolecular Engineering, University of Delaware, Newark, DE, USA. [7]Present address: SLAC National Accelerator Laboratory, Stanford University, Menlo Park, CA, USA. [8]These authors contributed equally: Ty Christoff-Tempesta, Yukio Cho. ✉e-mail: ortony@mit.edu

interactions with a solute[10,12]. Of note, hydration water can facilitate or inhibit the ability of dissolved species to interact with a surface to perform chemical processes[10,13]. Therefore, characterizing and leveraging the dynamics of a material's surface and its surrounding environment offers a critical pathway to mediating interfacial chemical events.

Small molecule assemblies formed by the spontaneous self-organization of amphiphiles in water represent a material class where interfacial behavior is critical to function[14–16]. The tunable surface chemistries and high surface areas hallmark of supramolecular assemblies offer promise for their use in a broad range of applications, including regenerative medicine[17], photonics[18], and water treatment[19]. Several reports have demonstrated the significant influence of internal conformational and solvation dynamics, on the properties of supramolecular assemblies[12,20–22], but surface and hydration dynamics at and above nanostructure surfaces remain largely unexplored in synthetic systems. Understanding the influence of interfacial behavior in this regime could enable new molecular design principles to enhance material performance.

Here, we characterize the interfacial dynamics of aramid amphiphile (AA) nanostructure surfaces and the impact of these dynamics on the nanomaterials' ability to remediate heavy metals from contaminated water (Fig. 1). AAs incorporate a triaramid structural domain to impart cohesive hydrogen bonding and a π-π stacking network to the internal domain of the resulting self-assembled nanostructures[23]. As a consequence, AA nanostructures demonstrate suppressed molecular exchange between assemblies and mechanical properties comparable to silk. Selecting the AA

design allows us to more readily isolate impacts from changing surface dynamics by minimizing dynamic instabilities pervasive in conventional supramolecular assemblies[20,23–25].

In this study, we incorporate oligo(ethylene glycol) (OEG) units of varying length between the AA structural domain and hydrophilic head group to vary surface dynamics (Fig. 1). OEG groups are well-established for their backbone flexibility and favorable interactions with water[26,27]. We hypothesize the incorporation of these groups into the molecular design of AAs will enhance surface dynamics and hydration, and consequently will improve water decontamination performance. We first characterize the self-assembly of the synthesized amphiphiles into internally organized nanostructures. Then, we co-assemble radical spin probes into the assembly surfaces to analyze molecular conformational dynamics using electron paramagnetic resonance (EPR) spectroscopy. Finally, we investigate the impacts of the differences in dynamic behavior among these assemblies on the nanostructures' ability to remediate heavy metal contaminants from the aqueous environment.

## Results and discussion
### Molecular design and self-assembly
Compounds (1)–(3) are AAs with anionic, heavy metal chelating head groups (derived from dodecane tetraacetic acid, DOTA) and either no inserted oligo(ethylene glycol) linker (compound (1)), an oligo(ethylene glycol) dimer (OEG$_2$, compound (2)), or an oligo(ethylene glycol) tetramer (OEG$_4$, compound (3)) between the AA structural domain and the head group (Fig. 1 and Supplementary Figs. 1–6). All compounds were analyzed by NMR spectroscopy and mass spectrometry, and

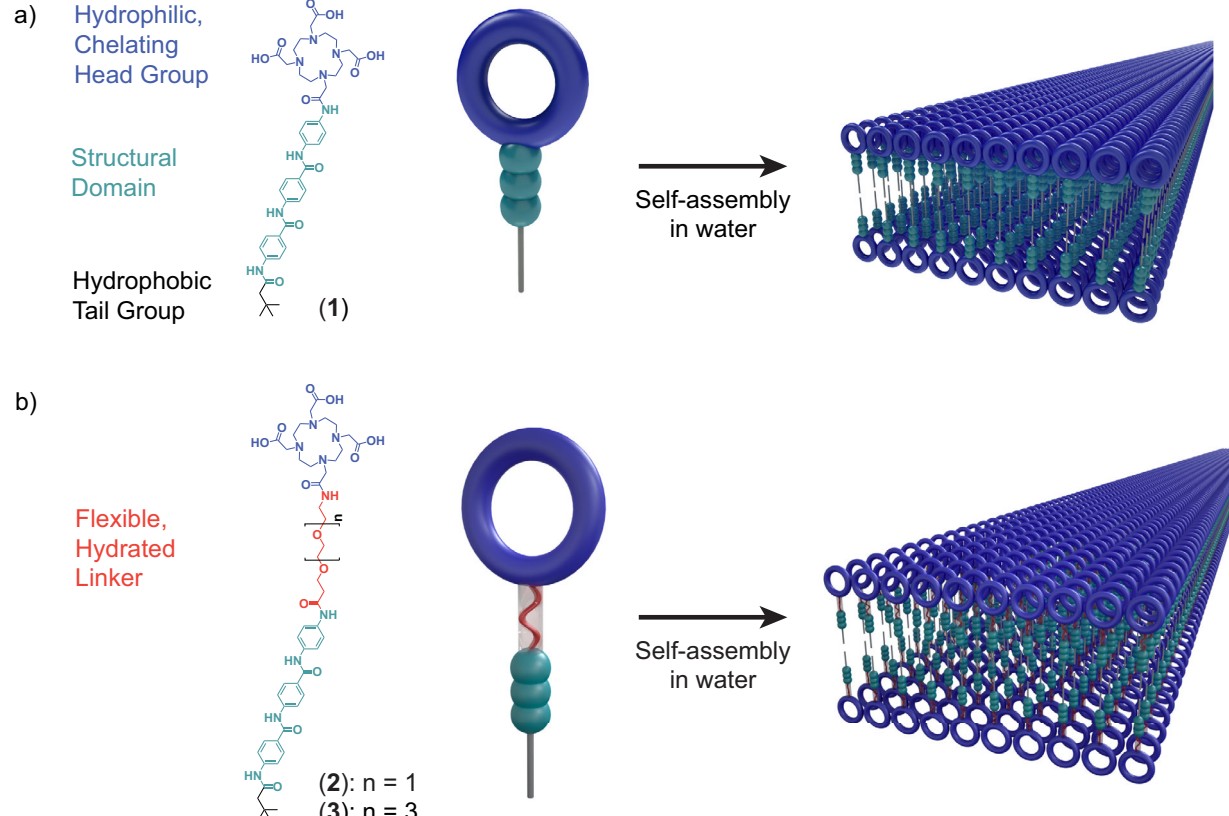

**Fig. 1 | Tunable surface chemistries characteristic of supramolecular assemblies enable control over surface dynamics and hydration. a** Prototypical aramid amphiphiles contain hydrophobic tail and hydrophilic head groups to assist self-assembly, and a structural domain to suppress dynamic exchange and enhance mechanical properties. For this study, a head group which is also capable of complexing heavy metals is chosen to assess the impact of surface dynamics on surface-mediated binding events. **b** The addition of oligo(ethylene glycol) linkers between the amphiphiles' hydrophobic and hydrophilic domains is hypothesized to enhance the local flexibility and hydration of the chelating head groups. Adapted with permission from ref. 19, Royal Society of Chemistry.

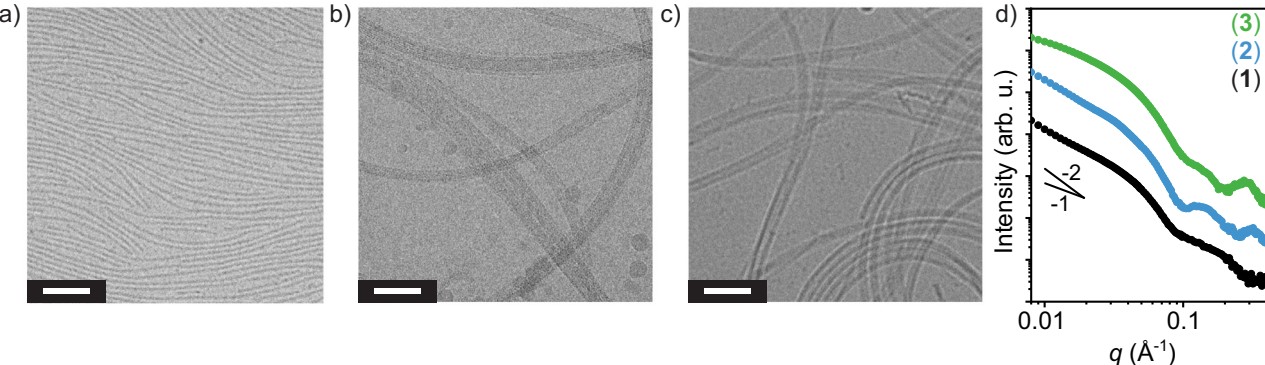

**Fig. 2 | Compounds (1)–(3) spontaneously self-assemble into microns-long nanoribbons in water.** The self-assembled nanostructures of compounds **a**, (**1**); **b**, (**2**); and (**c**), (**3**) are observed with cryogenic transmission electron microscopy (cryo-TEM). In all cases, the amphiphiles spontaneously assemble into a nanoribbon morphology. Some aggregation of compound (**2**) nanoribbons is observed. Scale bars, 50 nm. **d**, Synchrotron small angle X-ray scattering profiles of compound (**1**)–(**3**) assemblies support the observation of flexible, rod-like nanostructures observed in cryo-TEM, with low q regimes of slopes between −1 and −2. Curves are offset vertically for clarity.

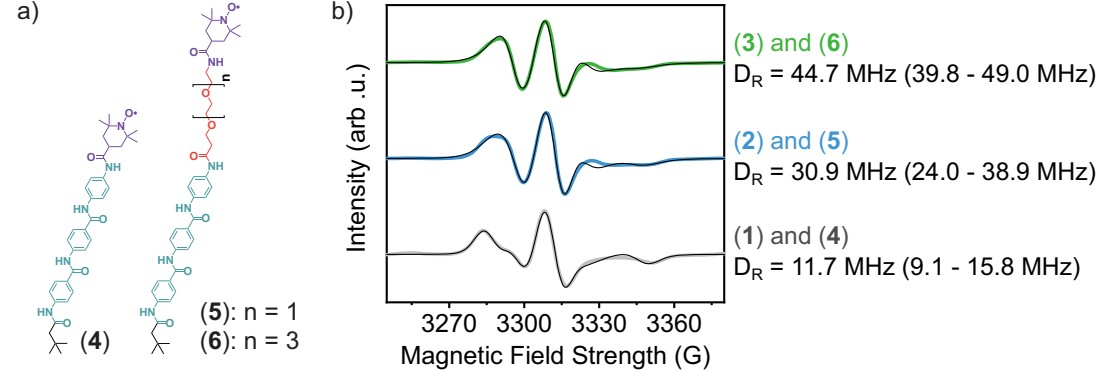

**Fig. 3 | Interfacial material and water dynamics are mediated through incorporation of flexible, hydrated surface linkers. a**, Compounds (**4**)–(**6**) are spin-labeled (purple) aramid amphiphiles that are co-assembled into compounds (**1**)–(**3**) (Fig. 1a), respectively, to probe dynamics near surface-tethered chelators. **b**, Electron paramagnetic resonance (EPR) spectroscopy of indicated co-assemblies reveals an over three-fold enhancement in the rotational diffusion constant of probes on nanoribbon surfaces from OEG$_4$-linker containing amphiphiles relative to those with no linker. Fits to each profile are shown with a black line and a 90% confidence interval to each reported diffusion constant is shown in parentheses. Curves are vertically offset for clarity.

synthesis and chemical characterization details are provided in the "Methods" section.

We observe compounds (**1**)–(**3**) spontaneously form ribbons with nanometer-scale cross-sections upon suspension in water via cryogenic transmission electron microscopy (cryo-TEM, Fig. 2a–c). Synchrotron small angle X-ray scattering (SAXS) further supports this finding, with all nanostructures demonstrating slopes between 1 and 2 at low $q$, indicative of flexible, rod-like structures (Fig. 2d)[28,29]. Cross-sectional analysis using higher $q$ data from SAXS is detailed later in the manuscript. In all cases, the nanoribbons extend microns in length. Infrared spectroscopy and wide-angle X-ray scattering analyses of the self-assembled nanostructures further indicate a cohesive hydrogen-bonding network is present in all assemblies (Supplementary Figs. 7 and 8)[23,30].

## Surface dynamics characterization

EPR techniques take advantage of site-directed spin labeling to quantify localized dynamics with sub-nanometer resolution[31]. By inserting radical nitroxide spin labels into a supramolecular structure, material dynamics at the spin label site can be captured over megahertz to gigahertz range of rotational diffusion rates ($D_R$, $10^6$–$10^9$ rad$^2$ s$^{-1}$)[32]. Typical spin labels minimally perturb the structure of molecular systems[12,33], and the high sensitivity of EPR techniques enables the use

of small amounts of spin labels to produce data[31]. Thus, EPR spectroscopy offers a route to representatively quantify the impacts of material flexibility and hydration dynamics on the dynamics of supramolecular nanostructure surfaces.

Compounds (**4**)–(**6**) were synthesized to probe localized dynamics at the sites of the chelating head groups through co-assembly at 5 mol% concentrations in nanoribbons of compounds (**1**)–(**3**), respectively (Fig. 3a, Supplementary Fig. 9). Compounds (**4**)–(**6**) are analogous AAs to compounds (**1**)–(**3**) in which the amphiphile head groups have been replaced with EPR spectroscopy-sensitive (2,2,6,6-tetramethylpiperidin-1-yl)oxyl (TEMPO) spin labels. We note that TEMPO spin-labeled AAs freely dissolved in a mixture of acetonitrile and water display three distinct peaks from isotropically tumbling nitroxide radicals, while those suspended in only water display a single, very broad peak arising from spin probe interactions indicative of molecular aggregation (Supplementary Fig. 10). In contrast, the broadened EPR spectra for mixtures of compounds (**4**)–(**6**) in compounds (**1**)–(**3**) in this study are well-described by a microscopic order/macroscopic disorder model (Supplementary Fig. 10)[34–36]. This implies that the spin-labeled AAs have been successfully co-assembled into the nanoribbons. With the inclusion and lengthening of an OEG linker in the head group's design, we find that rotational diffusion rates of the surface functionalities increase (Fig. 3b). Notably, $D_R$ nearly quadruples with

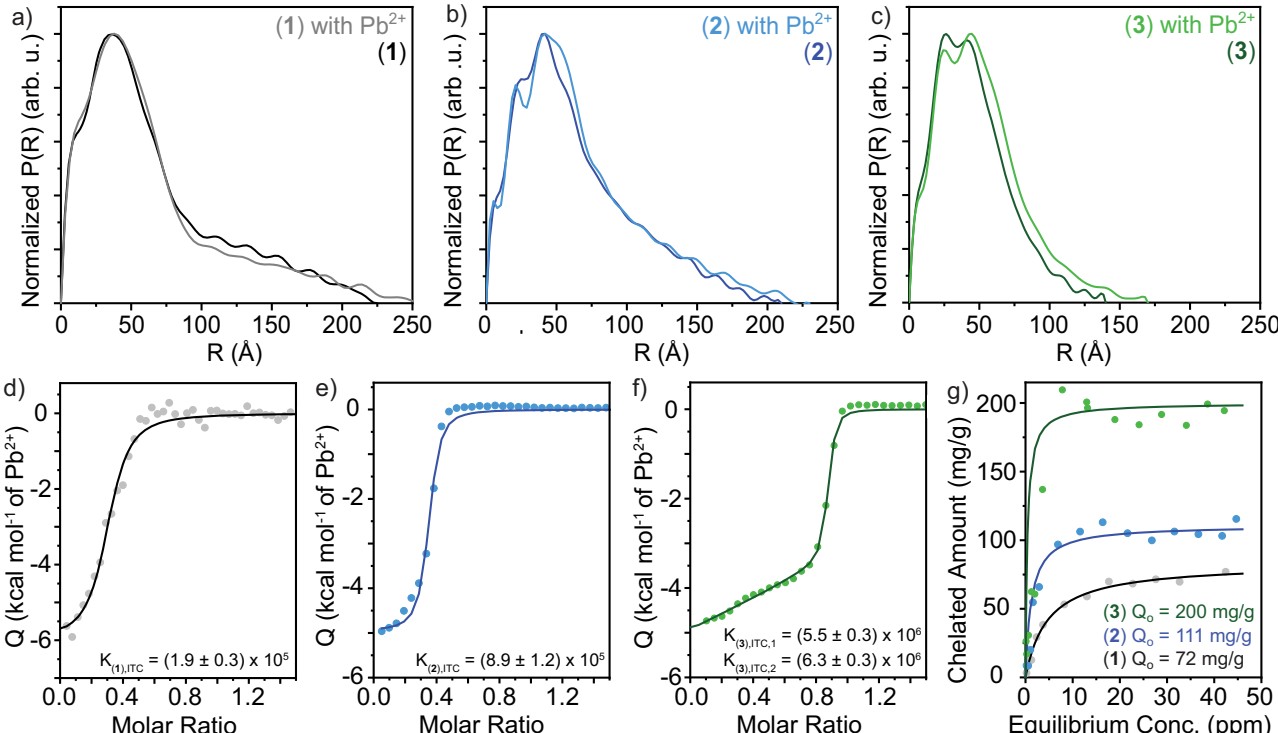

**Fig. 4 | Increasing surface dynamics, flexibility, and hydration enhances lead remediation. a–c** Pair distance distribution functions from small angle X-ray scattering profiles of compound (**1**)–(**3**) nanostructures imply the maintenance of internal organization upon the addition of $Pb^{2+}$ to solutions containing the nanoribbons through the conservation of curve shape and peak locations on the R axis. Nanoribbon thicknesses of ~7–9 nm are extracted from these profiles. **d–f** Isothermal titration calorimetry (ITC) measures the heat released from the complexation of $Pb^{2+}$ ions with tetraxetan head groups coating the supramolecular

assemblies' surfaces. ITC profiles of compound **d**, (**1**); **e**, (**2**); and **f**, (**3**) nanoribbons with $Pb^{2+}$ and their corresponding fits (darker lines) show increases in the equilibrium binding constant with the addition and extension of OEG linker units between amphiphile surface and internal domains. **g** Fitting adsorption isotherms of compound (**1**)–(**3**) nanoribbons with $Pb^{2+}$ to a Langmuir model (darker lines) reveals a significant enhancement in $Pb^{2+}$ remediation with enhanced surface dynamics. Notably, compound (**3**) nanoribbons saturate at ~200 mg $Pb^{2+}$ per gram of amphiphile.

the incorporation of $OEG_4$ between the internal and surface domains of the self-assembled nanostructures over those without a flexible linker. This enhancement may be attributed to both the flexibility of the OEG linkers and their capacity to preserve bulk hydration dynamics beyond the first hydration shell surrounding OEG moieties[37].

## Influence of interfacial behavior on surface binding events

We expect that modifying the interfacial dynamics of supramolecular nanostructures will have a significant impact on performing chemical events that harness surface interactions. Previously, we investigated the ability to employ aramid amphiphile nanoribbons in the removal of heavy metal ions from contaminated water[19]. These nanoribbons rely on surface-mediated interactions to complex dissolved heavy metal species with a chelating head group tethered to every amphiphile. In this study, we incorporate a tetraxetan head group onto compounds (**1**)–(**3**) due to its well-established affinity for binding to heavy metal ions[38], and probe the impact of modulating surface dynamics and hydration on the capture of $Pb^{2+}$ by compound (**1**)–(**3**) nanoribbons in water.

We first verify that the nanostructure and internal organization of compound (**1**)-(**3**) assemblies are preserved upon the addition of $Pb^{2+}$ to their aqueous environment through SAXS. We performed an indirect Fourier transform using GNOM on the SAXS scattering profiles of compound (**1**)–(**3**) nanostructures with and without stochiometric amounts of $Pb^{2+}$ to obtain pair distance distribution functions (PDDFs) of the nanostructure cross-sections in real space assuming monodisperse rods (Fig. 4a–c and Supplementary Figs. 11–13)[39]. This strategy allows us to obtain dimensional information from complex profiles arising from nanostructures with anisotropic dimensions and multiple regions with distinct scattering length densities[40,41]. From this analysis,

we find: compound (**1**), (**2**), and (**3**) nanoribbons with and without $Pb^{2+}$ are approximately 7.4, 8.4, and 8.8 nm thick, respectively. Notably, the internal organization of all nanoribbons remains similar before and after the addition of $Pb^{2+}$, as determined by the preservation of PDDF peak locations and shapes at R values centered around 12 and 37 Å for compound (**1**); 5, 23, and 42 Å for compound (**2**); and 7, 25, and 44 Å for compound (**3**). These features are hypothesized to correspond to approximately 25 Å-radii structural domains; 12 Å-thick DOTA head group layers; and 5 or 7 Å-thick $OEG_2$ or $OEG_4$ shells, respectively (Supplementary Fig. 13). The maximum cross-sectional dimensions of compound (**1**), (**2**), and (**3**) nanoribbons increase modestly from 22, 21, and 14 nm to 26, 23, and 17 nm, respectively, which may arise from an increase in the head group scattering length density with the chelation of $Pb^{2+}$, imply a slight increase in head group spacings upon the chelation of $Pb^{2+}$, or both. We note that the maximum cross-section dimensions are likely overestimated due to the flexibility of the ribbons and bundling caused by hydrogen bonding between head groups, as evidenced by the asymmetric tail of the PDDFs at high R values. The maintenance of nanostructure geometry with the addition of $Pb^{2+}$ is further supported by transmission electron microscopy (Supplementary Fig. 14).

Isothermal titration calorimetry (ITC), which measures the thermodynamics of binding interactions in solution, offers insight into the affinity of the nanoribbons for $Pb^{2+}$ by characterizing the stoichiometry and equilibrium binding constant ($K_{ITC}$) for complexes of the two species. We extract that the complex between tetraxetan head groups and $Pb^{2+}$ ions for compound (**1**) and (**2**) nanoribbons saturates near 50 mol% $Pb^{2+}$ (Fig. 4d, e), indicating a 2:1 head group:$Pb^{2+}$ complex stoichiometry consistent with a sandwich-like complex reported

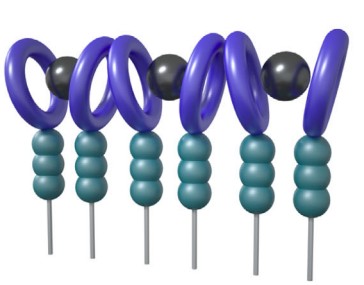
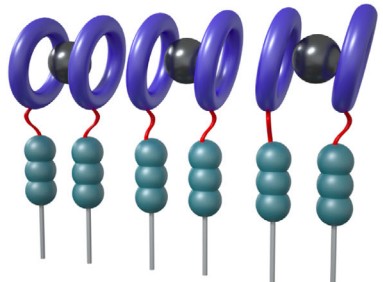
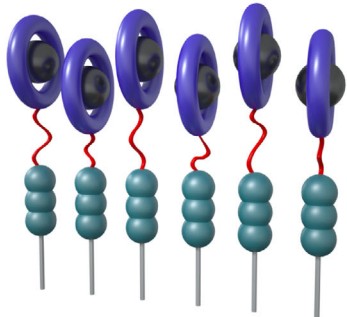

compound (**1**) nanoribbons          compound (**2**) nanoribbons          compound (**3**) nanoribbons

**Fig. 5 | Hypothesized nanoribbon surfaces illustrating how the addition and lengthening of oligo(ethylene glycol) linkers in the design of amphiphiles underlying self-assembled nanoribbons could enhance surface dynamics, flexibility, and spatial organization to mediate surface Pb²⁺ chelation.** The characterization reported in this manuscript suggests that the addition of a short OEG₂ linker between compound (**1**) and (**2**) nanoribbons enhances surface and interfacial water dynamics to improve Pb²⁺ binding but maintains chelating head groups in close proximity. The extension of this linker to OEG₄ in compound (**3**) nanoribbons combines enhancements in surface dynamics with additional spatial flexibility to enable each chelating head group to bind Pb²⁺ ions, resulting in a drastic improvement in Pb²⁺ remediation. We note that this illustration is a stylized interpretation of the complex formed between the DOTA chelating head groups and Pb²⁺ ions.

elsewhere[42]. In contrast, head groups tethered to compound (**3**) nanoribbons saturate near 100 mol% Pb²⁺ (Fig. 4f), indicating recovery of the 1:1 tetraxetan:Pb²⁺ complex observed in solution. We also observe notable enhancement in $K_{ITC}$ with increasing lengths of the OEG linker, and an order of magnitude increase in $K_{ITC}$ between compound (**1**) with no OEG linker and compound (**3**) with the OEG₄ linker (Fig. 4d–f). Two subsequent binding reactions are identified for compound (**3**) nanoribbons, which may indicate a switch between 2:1 sandwich-type and 1:1 head group:Pb²⁺ binding to accommodate more Pb²⁺ on the nanoribbon surfaces as the Pb²⁺ concentration increases.

Finally, we characterize the maximum amount of Pb²⁺ which can be removed from solution by each nanoribbon assembly with adsorption isotherms (Fig. 4g). The maximum saturation capacities ($Q_o$), reported as mg of Pb²⁺ removed from solution per g of amphiphile used, are determined by quantifying the plateau of the isotherms through fitting the adsorption behavior to a Langmuir model (Supplementary Fig. 15). We identify a modest improvement in Pb²⁺ removal in compound (**2**) assemblies that incorporate an OEG₂ linker into molecular design relative to assemblies of compound (**1**), and a larger enhancement in Pb²⁺ removal by compound (**3**) nanoribbons with an OEG₄ linker. These $Q_o$ values suggest that 700, 450, and 250 μg of compound (**1**), (**2**), and (**3**) nanoribbons, respectively, would be needed to remediate 1 L of 50 ppb (mass/vol) Pb²⁺-contaminated water. For context, the mass of a US penny is 2.5 g[43]; a penny's mass of compound (**3**) nanoribbons could treat up to 10,000 L of 50 ppb Pb²⁺-contaminated water.

By combining the results from EPR spectroscopy-based surface dynamics characterization with ITC and adsorption isotherms to investigate lead chelation, a clear trend emerges: incorporating longer OEG linkers into the AA design enhances surface dynamics and improves both the thermodynamic binding constant and the absolute chelation capacity. An intriguing observation merits attention: while $D_R$ increases substantially (2.6x) when transitioning from compound (**1**) to compound (**2**) nanoribbons, the increase in $D_R$ is more modest (1.4x) when transitioning from compound (**2**) to compound (**3**) nanoribbons. Surprisingly, enhancement of both the binding constant and the absolute chelation capacity is more pronounced between nanoribbons constructed of compound (**2**) and compound (**3**) than between nanoribbons of compound (**1**) and compound (**2**). These results suggest that while the chelation events are mediated by surface dynamics, dynamics alone is not fully determinant of surface behavior. Based on the totality of the material characterization, we hypothesize that the addition of an OEG₂ linker in compound (**2**) nanoribbons relative to compound (**1**) nanoribbons provides flexibility to the chelating groups,

promotes surface and interfacial water dynamics, and leads to the formation of more thermodynamically stable Pb²⁺ complexes (Fig. 5). In turn, we hypothesize that the incorporation of the longer OEG₄ linker between the internal and surface layers of compound (**3**) nanoribbons relative to the OEG₂ linker in compound (**2**) nanoribbons allows for concomitant spatial distribution of the chelating groups and enhancements in their dynamic behavior, leading to the significant improvement in the Pb²⁺-binding performance of the materials (Fig. 5).

Nature's pristine control over the dynamics of biological soft materials and their aqueous environments provides a powerful contention for leveraging flexibility and hydration in material design. In this report, we identified molecular design characteristics capable of enhancing interfacial material and water dynamics in supramolecular nanostructures designed for heavy metal remediation. We combined this control over dynamics with chemical design and the extraordinarily high surface areas characteristic of supramolecular assemblies to create nanostructures capable of remediating thousands of liters of 50 ppb Pb²⁺ contaminated water per gram of material. These results suggest that the conformational dynamics of the molecules that constitute a nanostructure, as well as the dynamics of surface water, can be harnessed to augment chemical events at the interface between a material and its aqueous environment.

## Methods

### Materials

Methyl 4-aminobenzoate (Sigma Aldrich, 98%), 3,3-dimethylbutyric acid (Sigma Aldrich, 98%), N-Boc-p-phenylenediamine (BPP, Sigma Aldrich, 97%), N-Boc-3-[2-(2-aminoethoxy)ethoxy]propionic acid (Ambeed Inc., 95%), Boc-15-amino-4,7,10,13-tetraoxapentadecanoic acid (Chem Impex. 95%), 2-(4,7,10-tris(2-tert-butoxy-2-oxoethyl) −1,4,7,10-tetraazacyclododecan-1-yl)acetic acid (DOTA-tris(t-Bu ester), AstaTech, 95%), 4-carboxy-2,2,6,6-tetramethylpiperidine 1-oxyl (4-carboxy-TEMPO, Sigma Aldrich, 97%), 1-ethyl-3-(3-dimethylaminopropyl)carbodiimide hydrochloride (EDC, TCI Chemicals, 98%), N,N'-Diisopropylcarbodiimide (DIC, Chem Impex. 99%), 4-dimethylaminopyridine (DMAP, TCI Chemicals, 99%), ethyl cyano(hydroxyimino)acetate (TCI Chemicals, 98%), lithium hydroxide monohydrate (LiOH·H₂O, Alfa Aesar, 98%), sodium bicarbonate (NaHCO₃, Alfa Aesar, 99%), hydrochloric acid (HCl, Alfa Aesar, 36%), sodium sulfate (Na₂SO₄, Fisher Scientific, 99%), magnesium sulfate (MgSO₄, J.T. Baker, anhydrous, 99%), sodium chloride (NaCl, Fisher Scientific, 99%), trifluoroacetic acid (TFA, Alfa Aesar, 99%), methanol (Fisher Scientific), acetonitrile (Fisher Scientific), methylene chloride (Fisher Scientific), N,N-dimethylformamide (dimethylformamide,

Fisher Scientific), and ethyl acetate (Fisher Scientific) were used as received without further purification.

## Synthesis and chemical characterization
Full synthesis and characterization details to obtain compounds (**1**)–(**6**) are provided in the Supplementary Information. In short, compounds (**1**)–(**6**) were synthesized using alternating carbodiimide-mediated amidation and standard deprotection reactions. $^1$H nuclear magnetic resonance (NMR) spectroscopy of samples in deuterated dimethylsulfoxide (DMSO-$d_6$) was conducted on a Bruker Avance III DPX 400. Molecular weights of synthesized compounds were investigated by matrix assisted laser desorption/ionization-time-of-flight mass spectrometry (MALDI-ToF MS) on a Bruker Omniflex instrument with a Reflectron accessory. The supernatant of a saturated α-cyano-4-hydroxycinnamic acid in 500:500:1 water:acetonitrile:TFA by volume solution was used as the MALDI-ToF matrix. MALDI-ToF samples were prepared by mixing amphiphile solutions with this matrix and SpheriCal Peptide Low (Polymer Factory) as an internal calibrant.

## Structural characterization
Self-assembled nanostructures were imaged by cryogenic transmission electron microscopy on a Talos Arctica G2 microscope set to a 200 kV accelerating voltage. Vitrified grids were prepared by pipetting 3 µL of 0.5 mg mL$^{-1}$ nanoribbon suspensions onto glow-discharged holey carbon grids (Quantifoil, 300 mesh, copper) in a FEI Vitrobot Mark IV at 100% humidity. Grids were then blotted for 4 s, plunged into liquid ethane, and preserved in liquid nitrogen.

Bulk structural characterization of nanostructure morphology was performed via small angle X-ray scattering (SAXS) at the LiX beamline of NSLS-II and Beamline 12-ID-B of the Advanced Photon Source (APS). SAXS at LiX/NSLS-II used 15.0 keV X-rays and a DECTRIS PILATUS3 1 M detector[44]. SAXS at 12-ID-B/APS used 13.3 keV X-rays and a DECTRIS EIGER 9 M detector. All SAXS was performed on 5 mg mL$^{-1}$ suspensions of molecular assemblies in quartz capillary tubes (Hampton Research, 2 mm diameter). The generation of 1D SAXS profiles for each sample and their background subtraction of a capillary filled with deionized water was performed with beamline software.

## Electron paramagnetic resonance spectroscopy
EPR spectra were collected at 298 K on a Bruker EMXplus spectrometer with the center field set at 3315 G and a 150 G sweep width. EPR spectroscopy samples were loaded into Teflon capillaries (1 mm inner diameter, 1.6 mm outer diameter, MSC Industrial Supply Co.) which were capped with Critoseal before analysis. Co-assemblies were prepared by mixing 5 mg mL$^{-1}$ solutions of compounds (**1**)–(**3**) dissolved in *N,N*-dimethylformamide (DMF) with their respective spin-labeled counterpart (**4**)–(**6**) dissolved in DMF with 5% NH$_4$OH (aq). These mixtures were held for 12 h at 80 °C to evaporate volatile components, lyophilized for 24 h to remove trace volatiles, suspended in deionized water to achieve an amphiphile concentration of 5 mg mL$^{-1}$, and bath sonicated for 1 h to produce nanoribbon co-assemblies. Exchange broadening is observed at spin label concentrations exceeding 10 mol% in the co-assemblies, so we selected 5 mol% spin label concentrations across all samples. Spectra were analyzed using the Chi-Squared Cluster Analysis (CSCA) spectral simulation toolkit[36], and the medoid is reported as the best representation of the rotational diffusion constant ($D_R$) from this fitting[36,45–47]. Further details are provided in the Supplementary Information.

## Characterization of heavy metal remediation
Aqueous solutions of Pb$^{2+}$ refer to lead (II) nitrate (Sigma-Aldrich) dissolved in deionized water.

Isothermal titration calorimetry (ITC) was performed on a MicroCal VP-ITC ultrasensitive titration calorimeter with 0.3 mM amphiphile and 3.0 mM Pb$^{2+}$ aqueous solutions. The background heat of dilution from injecting Pb$^{2+}$ into nanoribbon-free water was subtracted from all data. ITC experiments were performed in the absence of buffer due to the insolubility of lead species in most buffers. However, negligible signal was observed from the injection of water into nanoribbon solutions. Binding isotherms were captured at 25 °C and analyzed using instrument software.

Measurements of Pb$^{2+}$ concentrations to construct adsorption isotherms were taken on an Agilent 7900 inductively coupled plasma-mass spectrometer (ICP-MS). Samples were digested in a 2% hydrochloric acid / 2% nitric acid aqueous solution for analysis. The instrument was calibrated using a 10 ppm Pb standard (Ricca Chemical) and all samples were internally calibrated to a 10 ppm Rh standard (Sigma-Aldrich). To prepare samples for adsorption isotherm testing, aqueous mixtures with constant concentrations of compound (**1**)–(**3**) nanoribbons and variable concentrations of Pb$^{2+}$ were prepared, mixed, and equilibrated for 24 h. These solutions were then centrifuged for 5 min at 10,000 rcf and the supernatants were retained for analysis.

## Data availability
The data that support the findings of this study are available from the corresponding author upon request.

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

## Acknowledgements

We thank Ryan Allen for creating the nanoribbon images shown in Figs. 1 and 5 and Dr. Eszter Boros for discussions related to chelation complex formation. This material is based upon work supported by the National Science Foundation under Grant No. CHE-1945500. T.C-.T. acknowledges the support of the Hugh Hampton Young Fellowship and the Martin Family Society of Fellows for Sustainability. Y.C. acknowledges the support of the H.F. Taylor fellowship. This work made use of the MRSEC Shared Experimental Facilities at MIT, supported by the National Science Foundation under award number DMR-14-19807, and the MIT Department of Chemistry Instrumentation Facility. Specimens were prepared and imaged at the Automated Cryogenic Electron Microscopy Facility in MIT.nano on a Talos Arctica microscope, which was a gift from the Arnold and Mabel Beckman Foundation. The LiX beamline is part of the Center for BioMolecular Structure (CBMS), which is primarily supported by the National Institutes of Health, National Institute of General Medical Sciences (NIGMS) through a P30 Grant (P30GM133893), and by the DOE Office of Biological and Environmental Research (KP1605010). LiX also received additional support from NIH Grant S10 OD012331. As part of NSLS-II, a national user facility at Brookhaven National Laboratory, work performed at the CBMS is supported in part by the U.S. Department of Energy, Office of Science, Office of Basic Energy Sciences Program under contract number DE-SC0012704. This research used resources of the Advanced Photon Source, a U.S. Department of Energy (DOE) Office of Science user facility at Argonne National Laboratory and is based on research supported by the U.S. DOE Office of Science-Basic Energy Sciences, under Contract No. DE-AC02-06CH11357.

## Author contributions

T.C.-T. and Y.C. conceived and designed the experiments. T.C.-T., Y.C., and L.D.U. synthesized materials and performed chemical characterization. T.C.-T. and Y.C. performed MALDI TOF and FTIR of all samples. Y.C. performed TEM and Cryogenic TEM of all samples. Y.C. prepared samples for X-ray scattering and analyzed the X-ray scattering data. X.Z. performed X-ray scattering experiments. S.L.H. and L.D.P. calculated and plotted pair distance distribution functions from SAXS. T.C.-T., S.J.K., and L.D.U. performed EPR spectroscopy and analyzed the data. T.C.-T. and L.D.U. performed ITC and ICP-MS and completed formal analysis of the data. J.H.O., T.C.-T., and Y.C. co-wrote the manuscript. J.H.O. provided project administration, funding acquisition, and supervision. All authors discussed the results and commented on the manuscript.

## Competing interests

The authors declare no competing interests.
