## [Peer Review File · Nature Communications]

REVIEWER COMMENTS

Reviewer #1 (Remarks to the Author):

Christoff-Tempesta et al. wrote a very interesting manuscript on interfacial dynamics' effect on surface-mediated binding of metal ions. They found that adding OEG unit between the structural domain and the head group increases the rotational diffusion rate of the surface functionalities in nanoribbons; in other words, OEG makes the surface more flexible and dynamic. Later, they checked the performance of flexible surfaces on remediating metal(lead)-contaminated water. A more flexible surface showed a higher binding constant with metal ions. This work showed the importance of modifying interfaces with varying linker types, and sizes and has the potential to influence future research and industrial works on purifying contaminated water. I have a few concerns that I would like the authors to address before accepting it for publication.

1. Main claim of this paper is that flexibility and dynamics of the aramid amphiphiles increases the lead ion binding affinity. I am not fully convinced that increased dynamics is the main reason. DR values (figure 3(b)) indicate a significant increase in the rotational diffusion for compound (2) compared to compound (1). This dynamics enhancement is relatively weak when comparing compound (3) to compound (2). On the other hand, figure 4(d) shows chelated amount increased weakly (compounds 1 to 2) initially and then almost doubled (compounds 2 to 3) for the more flexible compound. This suggests that figure 4(d) is not a sole function of increased dynamics introduced by adding flexible OEG in the aramid amphiphiles. Discussion on how water dynamics is leading to this higher binding capacity can be helpful. In summary, the role of increased dynamics is unclear from the manuscript and needs more convincing information.
2. I think the flexibility of the linker is the most important factor for the increased binding of metal ions. Flexible OEG can help the surface to overcome the sandwich-like complex (page 8). Less flexibility can induce steric hindrance for the lead ions to adsorp. More flexible OEG can reduce the steric hindrance by spatially distributing the chelating group above the surface, creating more volume for the lead ions to adsorp.
3. The introduction section is too generic. The introduction should contain the preamble of what readers will learn from the "result and discussion" section, which needs to be added to the introduction. The system studied in the manuscript needs to be discussed or introduced in the introduction. The authors should also mention that they are interested in remediating lead ion contaminated water.

Reviewer #2 (Remarks to the Author):

The authors assembled nanoribbon structures using aramid amphiphiles with or without oligo(ethylene glycol) linkers. The nanostructures were examined using SAXS. The dynamics were characterized via EPR spectroscopy, supporting the faster dynamics with a longer OEG linker. The aramid amphiphile with a longer OEG linker was subsequently found to display a greatly elevated lead remediation performance. The work demonstrated the interesting dynamics features of nanoribbons and their potential application in heavy metal ions remediation. It is well written. However, I am not convinced that the dynamics alone is ascribed to the elevated lead remediation performance.

Major revisions:

1) The authors experimentally demonstrated that the dynamics follows the order of (1) < (2) < (3) (Fig. 3). However, I am not convinced that the difference in the dynamics plays the determinant role in the observed difference in the lead remediation (Fig. 4). Without the discussion of thermodynamics feature, dynamics is only one part of the whole story. Specifically, the ratios of 2:1 headgroup : Pb²⁺ for compounds (1, 2), but 1:1 for compound (3) are suggesting that the ionization of the headgroups in compounds (1, 2) is highly likely different from compound (3). The difference in the lead remediation performance between compounds (1) and (2) is probably related to the dynamics. Nevertheless, the difference between (1, 2) and (3) is likely primarily ascribed to the ionization of the headgroups (thermodynamics), instead of dynamics.

2) Two OEG length was investigated with the length of 2 and 4. Why not longer one?

Minor revisions:

1) (Abstract) “remediating thousands of liters of Pb²⁺-contaminated water with single grams of material”. Without the concentration of lead, this sentence is overselling.

2) (Page 3) “Small molecule supramolecular assemblies” needs to be rephased.

3) Use “oligo(ethylene glycol)” instead of “oligo-ethylene glycol”

4) (Page 5) “varying length between the AA structural domain and hydrophilic head group to systematically vary surface dynamics”. Only three compounds were investigated. Therefore, “systematically” is inaccurate.

5) NMR and MALDI-ToF MS were used to characterize the molecules synthesized here, but not provided in the manuscript and SI.

6) For the EPR spectroscopy, what are the ratios of the compounds (1-3) and the corresponding labelled counterpart (4-6)?

Reviewer #3 (Remarks to the Author):

This manuscript reports on the assembly of aramide amphiphiles that are decorated with OEG linkers of different length and carry a heavy metal chelating and how their interfacial dynamics influence the nanomaterials' heavy metal remediation performance. Depending on the molecular design, different Pb²⁺ removal performance was obtained.

The topic of the manuscript (study of interfacial dynamics of self-assembled materials) and the way the manuscript is presented is specific and, in my humble opinion, not suitable for a general readership or for a multidisciplinary journal. In addition, the same group has used almost identical amphiphiles forming the same type of self-assembled structure (ribbons) to recognize the same type of heavy atom (Pb²⁺; see "aramid amphiphile nanoribbons for the remediation of lead from contaminated water": *Environ. Sci.: Nano*, 2021, 8, 1536-1542). Thus, the main materials property is a slight enhancement in Pb²⁺ removal compared to previously reported systems, which is not a sufficiently strong argument to warrant publication in a top-quality multidisciplinary journal like *Nature Communications*. Further, thorough studies and deep understanding are needed in order to meet the technical quality criteria of the journal, which, in my humble opinion, is not the case. Given the specialized and incremental nature of this work, the lack of a new concept and the insufficient understanding of the system, my recommendation is to submit this manuscript to a specialized journal. Below, I highlight a number of points that the authors might consider for future resubmissions of this work:

- 1) Change in nanostructure morphology and properties upon addition of Pb is not analyzed.
- 2) Co-assembly is mentioned in the text, so the authors take for granted that it happens. However, no evidence on the co-assemblies is provided: no 2D NMR or spectroscopy studies or mathematical simulations to assess whether co-assembly occurs and, if so, what type of co-assembly and microstructure.
- 3) Molecular packing is not investigated. There is no influence of the molecular design on the aggregate topology (ribbon in all cases). The authors also mention "AAs incorporate a triaramid structural domain to impart a cohesive hydrogen bonding and π - π stacking network within the resulting self-assembled nanostructures". However, neither hydrogen bonding nor π - π stacking are examined, for instance by FTIR, NMR, UV/Vis, emission, etc.
- 4) The authors examine the binding of a specific metal cation (Pb²⁺), which was also investigated by them before. The system would be interesting if it could selectively recognize a specific metal ion, not just one that is already known. Molecular and supramolecular binders of heavy metal ions

have been widely developed for two decades and I would expect in this case a high level of sensitivity and selectivity to rival current systems.

5) How does the binding of Pb^{2+} occur? The authors mention the formation of a sandwich complex 2:1 for 1-2 and a 1:1 complex for 3, but the mechanism of binding by the ribbon structure is not explained. A cartoon would be helpful.

RESPONSE TO REVIEWS

Manuscript ID: NCOMMS-23-05266

Title: “Interfacial dynamics mediate surface binding events on supramolecular nanostructures”

Authors: Ty Christoff-Tempesta, Yukio Cho, Samuel J. Kaser, Linnaea D. Uliassi, Xiaobing Zuo, Shayna L. Hilburg, Lilo D. Pozzo, Julia H. Ortony

Reviewer #1 (Remarks to the Author):

Christoff-Tempesta et al. wrote a very interesting manuscript on interfacial dynamics' effect on surface-mediated binding of metal ions. They found that adding OEG unit between the structural domain and the head group increases the rotational diffusion rate of the surface functionalities in nanoribbons; in other words, OEG makes the surface more flexible and dynamic. Later, they checked the performance of flexible surfaces on remediating metal(lead)-contaminated water. A more flexible surface showed a higher binding constant with metal ions. This work showed the importance of modifying interfaces with varying linker types, and sizes and has the potential to influence future research and industrial works on purifying contaminated water. I have a few concerns that I would like the authors to address before accepting it for publication.

1. Main claim of this paper is that flexibility and dynamics of the aramid amphiphiles increases the lead ion binding affinity. I am not fully convinced that increased dynamics is the main reason. DR values (figure 3(b)) indicate a significant increase in the rotational diffusion for compound (2) compared to compound (1). This dynamics enhancement is relatively weak when comparing compound (3) to compound (2). On the other hand, figure 4(d) shows chelated amount increased weakly (compounds 1 to 2) initially and then almost doubled (compounds 2 to 3) for the more flexible compound. This suggests that figure 4(d) is not a sole function of increased dynamics introduced by adding flexible OEG in the aramid amphiphiles. Discussion on how water dynamics is leading to this higher binding capacity can be helpful. In summary, the role of increased dynamics is unclear from the manuscript and needs more convincing information.

We thank Reviewer 1 for this comment, which was echoed by Reviewers 2 and 3, and have updated the manuscript significantly as a result. We agree that modifying the nanofiber surface through the addition of the OEG linkers likely causes a series of complementary effects, including modifying dynamics among other properties, that led to enhanced binding performance. As a result, we have added discussion to the manuscript and softened claims based on dynamics alone. Our added discussion is copied below for reference:

By combining the results from EPR spectroscopy-based surface dynamics characterization with ITC and adsorption isotherms to investigate lead chelation, a clear trend emerges: incorporating longer OEG linkers into the AA design enhances surface dynamics and improves both the thermodynamic binding constant and the absolute chelation capacity. An intriguing observation merits attention: while D_R increases substantially (2.6x) when transitioning from compound (1) to compound (2) nanoribbons, the increase in D_R is more modest (1.4x) when transitioning from compound (2) to compound (3) nanoribbons. Surprisingly, enhancement of both the binding constant and the absolute chelation capacity is more pronounced between nanoribbons constructed of compound (2) and compound (3) than between nanoribbons of compound (1) and compound (2). These results suggest that while the chelation events are mediated by surface dynamics, dynamics alone is not fully determinant of surface behavior. Based on the totality of the material characterization, we hypothesize that the addition of an OEG₂ linker in compound (2) nanoribbons relative to compound (1) nanoribbons provides flexibility to the chelating groups, promotes surface and interfacial water dynamics, and leads to the formation of more thermodynamically stable Pb²⁺ complexes (Figure 5). In turn, we hypothesize that the incorporation of the longer OEG₄ linker between the internal and surface layers of compound (3) nanoribbons relative to the OEG₂ linker in compound (2) nanoribbons allows for concomitant spatial distribution of the chelating groups and enhancements in their dynamic behavior, leading to the significant improvement in the Pb²⁺-binding performance of the materials (Figure 5).

Fig. 5 | Hypothesized nanoribbon surfaces illustrating how the addition and lengthening of oligo(ethylene glycol) linkers in the design of amphiphiles underlying self-assembled nanoribbons enhance surface dynamics, flexibility, and spatial organization to mediate surface Pb²⁺ chelation. The characterization reported in this manuscript suggests that the addition of a short OEG₂ linker between compound (1) and (2) nanoribbons enhances surface and interfacial water dynamics to improve Pb²⁺ binding but maintains chelating head groups in close proximity. The extension of this linker to OEG₄ in compound (3) nanoribbons combines enhancements in surface dynamics with additional spatial flexibility to enable each chelating head group to bind Pb²⁺ ions, resulting in a drastic improvement in Pb²⁺ remediation.

2. I think the flexibility of the linker is the most important factor for the increased binding of metal ions. Flexible OEG can help the surface to overcome the sandwich-like complex (page 8). Less flexibility can induce steric hindrance for the lead ions to adsorp. More flexible OEG can reduce the steric hindrance by spatially distributing the chelating group above the surface, creating more volume for the lead ions to adsorp.

We agree with this assessment and appreciate that it was not clearly expressed in the originally submitted version of the manuscript. We have updated the manuscript with additional discussion and an added schematic to address this consideration; this change is copied in the previous response.

3. The introduction section is too generic. The introduction should contain the preamble of what readers will learn from the "result and discussion" section, which needs to be added to the introduction. The system studied in the manuscript needs to be discussed or introduced in the introduction. The authors should also mention that they are interested in remediating lead ion contaminated water.

We have moved the beginning of the Results and Discussion section, which previews the contents of the paper, into the Introduction to improve the flow of the paper and added text to provide additional context. This updated region of the manuscript is copied below for reference:

Here, we characterize the interfacial dynamics of aramid amphiphile (AA) nanostructure surfaces and the impact of these dynamics on the nanomaterials' ability to remediate heavy metals from contaminated water (Figure 1). AAs incorporate a triaramid structural domain to impart cohesive hydrogen bonding and a π - π stacking network to the internal domain of the resulting self-assembled nanostructures²³. As a consequence, AA nanostructures demonstrate suppressed molecular exchange between assemblies and mechanical properties comparable to silk. Selecting the AA design allows us to more readily isolate impacts from changing surface dynamics by minimizing dynamic instabilities pervasive in conventional supramolecular assemblies^{20,23-25}.

In this study, we incorporate oligo(ethylene glycol) (OEG) units of varying length between the AA structural domain and hydrophilic head group to vary surface dynamics (Figure 1). OEG groups are well-established for their backbone flexibility and favorable interactions with water^{26,27}. We hypothesize the incorporation of these groups into the molecular design of AAs will enhance surface dynamics and hydration, and consequently will improve water decontamination performance. We first characterize the self-assembly of the synthesized amphiphiles into internally organized nanostructures. Then, we co-assemble radical spin probes into the assembly surfaces to analyze molecular conformational dynamics using electron paramagnetic resonance (EPR) spectroscopy. Finally, we investigate the impacts of the differences in dynamic behavior among these assemblies on the nanostructures' ability to remediate heavy metal contaminants from the aqueous environment.

Reviewer #2 (Remarks to the Author):

The authors assembled nanoribbon structures using aramid amphiphiles with or without oligo(ethylene glycol) linkers. The nanostructures were examined using SAXS. The dynamics were characterized via EPR spectroscopy, supporting the faster dynamics with a longer OEG linker. The aramid amphiphile with a longer OEG linker was subsequently found to display a greatly elevated lead remediation performance. The work demonstrated the interesting dynamics features of nanoribbons and their potential application in heavy metal ions remediation. It is well written. However, I am not convinced that the dynamics alone is ascribed to the elevated lead remediation performance.

Major revisions:

1) The authors experimentally demonstrated that the dynamics follows the order of (1) < (2) < (3) (Fig. 3). However, I am not convinced that the difference in the dynamics plays the determinant role in the observed difference in the lead remediation (Fig. 4). Without the discussion of thermodynamics feature, dynamics is only one part of the whole story. Specifically, the ratios of 2:1 headgroup : Pb²⁺ for compounds (1, 2), but 1:1 for compound (3) are suggesting that the ionization of the headgroups in compounds (1, 2) is highly likely different from compound (3). The difference in the lead remediation performance between compounds (1) and (2) is probably related to the dynamics. Nevertheless, the difference between (1, 2) and (3) is likely primarily ascribed to the ionization of the headgroups (thermodynamics), instead of dynamics.

We appreciate this comment, which is echoed by Reviewers 1 and 3, and agree that dynamics forms a part of a larger story. We have updated the manuscript in a few ways to address this comment.

(1) We originally reported the rotational diffusion constant (D_R) as a logarithmic value based on historical EPR conventions. However, we feel that the display of the value in this way may overemphasize the perceived difference between the D_R values. Therefore, we are now representing the D_R values in integer format to enhance their accessibility to the broader audience of *Nat. Comm.* For reference, these values and their 90% confidence intervals (in parentheses) are copied below:

- Compound 1+4 (no OEG linker): 11.7 MHz (90% CI: 9.1 - 15.8 MHz)
- Compound 2+5 (OEG₂ linker): 30.9 MHz (90% CI: 24.0-38.9 MHz)
- Compound 3+6 (OEG₄ linker): 44.7 MHz (90% CI: 39.8 - 49.0 MHz)

(2) We agree that there are synergistic effects arising from the addition and lengthening of the OEG linkers that are not captured by dynamics alone. We appreciate that this was not well-discussed before and have added the following text and figure to the manuscript:

By combining the results from EPR spectroscopy-based surface dynamics characterization with ITC and adsorption isotherms to investigate lead chelation, a clear trend emerges: incorporating longer OEG linkers into the AA design enhances surface dynamics and improves both the thermodynamic binding constant and the absolute chelation capacity. An intriguing observation merits attention: while D_R increases substantially (2.6x) when transitioning from compound (1) to compound (2) nanoribbons, the increase in D_R is more modest (1.4x) when transitioning from compound (2) to compound (3) nanoribbons. Surprisingly, enhancement of both the binding constant and the absolute chelation capacity is more pronounced between nanoribbons constructed of compound (2) and compound (3) than between nanoribbons of compound (1) and compound (2). These results suggest that while the chelation events are mediated by surface dynamics, dynamics alone is not fully determinant of surface behavior. Based on the totality of the material characterization, we hypothesize that the addition of an OEG₂ linker in compound (2) nanoribbons relative to compound (1) nanoribbons provides flexibility to the chelating groups, promotes surface and interfacial water dynamics, and leads to the formation of more thermodynamically stable Pb²⁺ complexes (Figure 5). In turn, we hypothesize that the incorporation of the longer OEG₄ linker between the internal and surface layers of compound (3) nanoribbons relative to the OEG₂ linker in compound (2) nanoribbons allows for concomitant spatial distribution of the chelating groups and enhancements in their dynamic behavior, leading to the significant improvement in the Pb²⁺-binding performance of the materials (Figure 5).

Fig. 5 | Hypothesized nanoribbon surfaces illustrating how the addition and lengthening of oligo(ethylene glycol) linkers in the design of amphiphiles underlying self-assembled nanoribbons enhance surface dynamics, flexibility, and spatial organization to mediate surface Pb²⁺ chelation. The characterization reported in this manuscript suggests that the addition of a short OEG₂ linker between compound (1) and (2) nanoribbons enhances surface and interfacial water dynamics to improve Pb²⁺ binding but maintains chelating head groups in close proximity. The extension of this linker to OEG₄ in compound (3) nanoribbons combines enhancements in surface dynamics with additional spatial flexibility to enable each chelating head group to bind Pb²⁺ ions, resulting in a drastic improvement in Pb²⁺ remediation.

2) Two OEG length was investigated with the length of 2 and 4. Why not longer one?

We agree that investigations into AAs with even longer OEG linkers would be interesting to probe a broader range of surface dynamics. However, as the length of the OEG linker increases, the precision of the length decreases because the dispersity of the number of ethylene glycol repeat units in commercially available compounds becomes substantial. In response to this comment, we synthesized an additional aramid amphiphile with an OEG₁₂ linker between the internal and chelating domains. The OEG₁₂ linker was chosen in order to probe whether differences in dynamics arise from much longer linker lengths compared to OEG₄, which is the longest linker reported on in this manuscript and is denoted as compound (3). We found that nanostructures assembled from this OEG₁₂-containing AA performed similarly in ITC experiments as compound (3) nanostructures, so we are confident that the range of amphiphiles reported in the manuscript probe the range of dynamic behavior of interest. Importantly, a functionalizable OEG₁₂ molecule was not available to synthesize a counterpart TEMPO-linked AA for EPR, so we are unable to perform site-directed dynamics characterization on the surfaces of nanostructures from OEG₁₂-linked AAs. Therefore, we decided to focus our efforts on a more thorough characterization of assemblies within the bounds reported in the manuscript.

Minor revisions:

1) (Abstract) “remediating thousands of liters of Pb²⁺-contaminated water with single grams of material”. Without the concentration of lead, this sentence is overselling.

This is an important detail; thank you for catching it. We have updated the abstract and concluding paragraph to include the Pb²⁺ concentration (50 ppb).

2) (Page 3) “Small molecule supramolecular assemblies” needs to be rephased.

We have shortened this phrase to “small molecule assemblies.”

3) Use “oligo(ethylene glycol)” instead of “oligo-ethylene glycol”

We have updated this terminology throughout the manuscript.

4) (Page 5) “varying length between the AA structural domain and hydrophilic head group to systematically vary surface dynamics”. Only three compounds were investigated. Therefore, “systematically” is inaccurate.

We have removed this word from the manuscript.

5) NMR and MALDI-ToF MS were used to characterize the molecules synthesized here, but not provided in the manuscript and SI.

Thank you for catching this. We have corrected the SI to include the chemical characterization.

6) For the EPR spectroscopy, what are the ratios of the compounds (1-3) and the corresponding labelled counterpart (4-6)?

We appreciate the need to clarify this in the main text of the manuscript; it was previously only located in the Methods section. The compound (4-6) spin labels were added at 5 mol% relative to compounds (1-3). This detail has been added to the corresponding section of the Results and Discussion.

Reviewer #3 (Remarks to the Author):

This manuscript reports on the assembly of aramide amphiphiles that are decorated with OEG linkers of different length and carry a heavy metal chelating and how their interfacial dynamics influence the nanomaterials' heavy metal remediation performance. Depending on the molecular design, different Pb²⁺ removal performance was obtained.

The topic of the manuscript (study of interfacial dynamics of self-assembled materials) and the way the manuscript is presented is specific and, in my humble opinion, not suitable for a general readership or for a multidisciplinary journal. In addition, the same group has used almost identical amphiphiles forming the same type of self-assembled structure (ribbons) to recognize the same type of heavy atom (Pb²⁺; see “aramid amphiphile nanoribbons for the remediation of lead from contaminated water”: Environ. Sci.: Nano, 2021, 8, 1536-1542). Thus, the main materials property is a slight enhancement in Pb²⁺ removal compared to previously reported systems, which is not a sufficiently strong argument to warrant publication in a top-quality multidisciplinary journal like Nature Communications. Further, thorough studies and deep understanding are needed in order to meet the technical quality criteria of the journal, which, in my humble opinion, is not the case. Given the specialized and incremental nature of this work, the lack of a new concept and the insufficient understanding of the system, my recommendation is to submit this manuscript to a specialized journal. Below, I highlight a number of points that the authors might consider for future resubmissions of this work:

We thank Reviewer 3 for their suggestions and appreciate the opportunity to address them to highlight the impact of our manuscript, emphasizing the fundamental advance in tying quantitative measurements of active site dynamics on self-assembled systems to their performance. We note that this finding has significant implications for all soft matter systems in which interfacial

behavior determines function, and selected water treatment as an exemplary, meaningful application space to examine the impact of our discovery. We note that we respond in depth to these remarks in Reviewer 3, Comment 4.

1) Change in nanostructure morphology and properties upon addition of Pb is not analyzed.

In combination with Reviewer 3, comment 3, we have added new TEM and X-ray experiments to the paper that verify the nanostructure morphology is maintained throughout the addition of Pb^{2+} . These additions are copied below alongside the updated Figure 4:

We first verify that the nanostructure and internal organization are preserved upon the addition of Pb^{2+} to their aqueous environment through SAXS. We performed an indirect Fourier transform using GNOM on the scattering profiles of compound (1) – (3) nanostructures with and without stoichiometric amounts of Pb^{2+} to obtain pair distance distribution functions (PDDFs) of the nanostructure cross-sections in real space assuming monodisperse rods (Figure 4a-c, Supplementary Figures 10-12).³⁹ This strategy allows us to obtain dimensional information from complex profiles arising from nanostructures with anisotropic dimensions and multiple regions with distinct scattering length densities.^{40,41} From this analysis, we find: compound (1), (2), and (3) nanoribbons with and without Pb^{2+} are approximately 7.4, 8.4, and 8.8 nm thick, respectively. Notably, the internal organization of all nanoribbons remains similar before and after the addition of Pb^{2+} , as determined by the preservation of PDDF peak locations and shapes at R values centered around 12 and 37 Å for compound (1); 5, 23, and 42 Å for compound (2); and 7, 25, and 44 Å for compound (3). These features are hypothesized to correspond to approximately 25 Å-radii structural domains; 12 Å-thick DOTA head group layers; and 5 or 7 Å-thick OEG₂ or OEG₄ shells, respectively (Supplementary Figure 12). The maximum cross-sectional dimensions of compound (1), (2), and (3) nanoribbons increase modestly from 22, 21, and 14 nm to 26, 23, and 17 nm, respectively, which may arise from an increase in the head group scattering length density with the chelation of Pb^{2+} , imply a slight increase in head group spacings upon the chelation of Pb^{2+} , or both. We note that the maximum cross-section dimensions are likely overestimated due to the flexibility of the ribbons and bundling caused by hydrogen bonding between head groups, as evidenced by the asymmetric tail of the PDDFs at high R values. The maintenance of nanostructure geometry with the addition of Pb^{2+} is further supported by transmission electron microscopy (Supplementary Figure 13).

[Continued on next page]

Fig. 4 | Increasing surface dynamics, flexibility, and hydration enhances lead remediation. a-c, Pair distance distribution functions from small angle X-ray scattering profiles of compound (1)–(3) nanostructures imply the maintenance of internal organization upon the addition of Pb²⁺ to solutions containing the nanoribbons through the conservation of curve shape and peak locations on the R axis. Nanoribbon thicknesses of ~7-9 nm are extracted from these profiles. Curves are offset for clarity. d-f, Isothermal titration calorimetry (ITC) measures the heat released from the complexation of Pb²⁺ ions with tetraacetate head groups coating the supramolecular assemblies' surfaces. ITC profiles of compound d, (1); e, (2); and f, (3) nanoribbons with Pb²⁺ and their corresponding fits (darker lines) show increases in the equilibrium binding constant with the addition and extension of OEG linker units between amphiphile surface and internal domains. g, Fitting adsorption isotherms of compound (1)–(3) nanoribbons with Pb²⁺ to a Langmuir model (darker lines) reveals a significant enhancement in Pb²⁺ remediation with enhanced surface dynamics. Notably, compound (3) nanoribbons saturate at approx. 200 mg Pb²⁺ per gram of amphiphile.

2) Co-assembly is mentioned in the text, so the authors take for granted that it happens. However, no evidence on the co-assemblies is provided: no 2D NMR or spectroscopy studies or mathematical simulations to assess whether co-assembly occurs and, if so, what type of co-assembly and microstructure.

Thank you for highlighting this important point. We conducted additional experiments to validate the co-assembly and have included the results in the Supplementary Information, with clarifications in the manuscript. These additions are copied below:

[Manuscript] We note that TEMPO spin-labeled AAs freely dissolved in water display three distinct peaks from isotopically tumbling nitroxide radicals (Supplementary Figure 9). In contrast, the broadened EPR spectra for mixtures of compounds (4) - (6) in compounds (1) - (3) in this study are well-described by a microscopic order/macroscopic

disorder model (Supplementary Figure 9)³⁴⁻³⁶. This implies that the spin-labeled AAs have been successfully co-assembled into the nanoribbons.

Supplementary Figure 9. Representative EPR spectra of aqueous suspensions of compound (3), 5 mol% of compound (6) in compound (3), and compound (6). The mixture of compound (3) in (6) exhibits a macroscopic order-microscopic disorder-like lineshape, implying co-assembly. In contrast, aqueous suspensions containing only compound (3) display no EPR signal. Further, aqueous suspensions of compound (6) alone reveal a free tumbling lineshape.

3) Molecular packing is not investigated. There is no influence of the molecular design on the aggregate topology (ribbon in all cases). The authors also mention “AAs incorporate a triamid structural domain to impart a cohesive hydrogen bonding and π - π stacking network within the resulting self-assembled nanostructures”. However, neither hydrogen bonding nor π - π stacking are examined, for instance by FTIR, NMR, UV/Vis, emission, etc.

We have added new X-ray scattering, TEM, and FTIR experiments and analysis to the manuscript to address this comment. These changes are copied below alongside the updated Figure 4:

[In the *Molecular Design and Self-Assembly* section] Synchrotron small angle X-ray scattering (SAXS) further supports this finding, with all nanostructures demonstrating slopes between 1 and 2 at low q , indicative of flexible, rod-like structures (Figure 2d).^{28,29} Cross-sectional analysis using higher q data from SAXS is detailed later in the manuscript. In all cases, the nanoribbons extend microns in length. Infrared spectroscopy and wide

angle X-ray scattering analyses of the self-assembled nanostructures further indicate a cohesive hydrogen-bonding network is present in all assemblies (Supplementary Figures 7-8).^{23,30}

[In the *Influence of Interfacial Behavior on Surface Performance* section] We first verify that the nanostructure and internal organization are preserved upon the addition of Pb^{2+} to their aqueous environment through SAXS. We performed an indirect Fourier transform using GNOM on the scattering profiles of compound (1) – (3) nanostructures with and without stoichiometric amounts of Pb^{2+} to obtain pair distance distribution functions (PDDFs) of the nanostructure cross-sections in real space assuming monodisperse rods (Figure 4a-c, Supplementary Figures 10-12).³⁹ This strategy allows us to obtain dimensional information from complex profiles arising from nanostructures with anisotropic dimensions and multiple regions with distinct scattering length densities.^{40,41} From this analysis, we find: compound (1), (2), and (3) nanoribbons with and without Pb^{2+} are approximately 7.4, 8.4, and 8.8 nm thick, respectively. Notably, the internal organization of all nanoribbons remains similar before and after the addition of Pb^{2+} , as determined by the preservation of PDDF peak locations and shapes at R values centered around 12 and 37 Å for compound (1); 5, 23, and 42 Å for compound (2); and 7, 25, and 44 Å for compound (3). These features are hypothesized to correspond to approximately 25 Å-radii structural domains; 12 Å-thick DOTA head group layers; and 5 or 7 Å-thick OEG₂ or OEG₄ shells, respectively (Supplementary Figure 12). The maximum cross-sectional dimensions of compound (1), (2), and (3) nanoribbons increase modestly from 22, 21, and 14 nm to 26, 23, and 17 nm, respectively, which may arise from an increase in the head group scattering length density with the chelation of Pb^{2+} , imply a slight increase in head group spacings upon the chelation of Pb^{2+} , or both. We note that the maximum cross-section dimensions are likely overestimated due to the flexibility of the ribbons and bundling caused by hydrogen bonding between head groups, as evidenced by the asymmetric tail of the PDDFs at high R values. The maintenance of nanostructure geometry with the addition of Pb^{2+} is further supported by transmission electron microscopy (Supplementary Figure 13).

[Continued on next page]

Fig. 4 | Increasing surface dynamics, flexibility, and hydration enhances lead remediation. a-c, Pair distance distribution functions from small angle X-ray scattering profiles of compound (1)–(3) nanostructures imply the maintenance of internal organization upon the addition of Pb^{2+} to solutions containing the nanoribbons through the conservation of curve shape and peak locations on the R axis. Nanoribbon thicknesses of ~ 7 – 9 nm are extracted from these profiles. Curves are offset for clarity. d-f, Isothermal titration calorimetry (ITC) measures the heat released from the complexation of Pb^{2+} ions with tetraacetan head groups coating the supramolecular assemblies' surfaces. ITC profiles of compound d, (1); e, (2); and f, (3) nanoribbons with Pb^{2+} and their corresponding fits (darker lines) show increases in the equilibrium binding constant with the addition and extension of OEG linker units between amphiphile surface and internal domains. g, Fitting adsorption isotherms of compound (1)–(3) nanoribbons with Pb^{2+} to a Langmuir model (darker lines) reveals a significant enhancement in Pb^{2+} remediation with enhanced surface dynamics. Notably, compound (3) nanoribbons saturate at approx. 200 mg Pb^{2+} per gram of amphiphile.

4) The authors examine the binding of a specific metal cation (Pb^{2+}), which was also investigated by them before. The system would be interesting if it could selectively recognize a specific metal ion, not just one that is already known. Molecular and supramolecular binders of heavy metal ions have been widely developed for two decades and I would expect in this case a high level of sensitivity and selectivity to rival current systems.

Self-assembled nanostructures have primarily been investigated for biological and biomedical applications due to their transient intermolecular interactions that give rise to dynamic exchange processes. Previously, we demonstrated that the design of self-assembled nanostructures with strong intermolecular interactions could expand the application space of supramolecular materials to water treatment. In contrast, this manuscript shows a direct relationship between the nanoscale motion of an active group tethered to a self-assembled nanostructure and its potency. This effect

is important for any nanostructure in water for any application, including those motivated by biology, energy, catalysis, and the environment. We feel that this contribution is significant and of interest to a broad audience and enabled by performing quantitative EPR at the active site to measure dynamics. We chose to examine heavy metal remediation in this study as an example application space that could be impacted by our discovery based on our expertise in the area but consider the fundamental advancement to be a broad consideration for all soft matter systems where interfacial behavior is critical to function.

5) How does the binding of Pb²⁺ occur? The authors mention the formation of a sandwich complex 2:1 for 1-2 and a 1:1 complex for 3, but the mechanism of binding by the ribbon structure is not explained. A cartoon would be helpful.

We appreciate that this aspect of the system was not well-described in the original submission of the manuscript and is important to the function of the system. We have added further context and a cartoon, as suggested, to the paper, copied below:

By combining the results from EPR spectroscopy-based surface dynamics characterization with ITC and adsorption isotherms to investigate lead chelation, a clear trend emerges: incorporating longer OEG linkers into the AA design enhances surface dynamics and improves both the thermodynamic binding constant and the absolute chelation capacity. An intriguing observation merits attention: while D_R increases substantially (2.6x) when transitioning from compound (1) to compound (2) nanoribbons, the increase in D_R is more modest (1.4x) when transitioning from compound (2) to compound (3) nanoribbons. Surprisingly, enhancement of both the binding constant and the absolute chelation capacity is more pronounced between nanoribbons constructed of compound (2) and compound (3) than between nanoribbons of compound (1) and compound (2). These results suggest that while the chelation events are mediated by surface dynamics, dynamics alone is not fully determinant of surface behavior. Based on the totality of the material characterization, we hypothesize that the addition of an OEG₂ linker in compound (2) nanoribbons relative to compound (1) nanoribbons provides flexibility to the chelating groups, promotes surface and interfacial water dynamics, and leads to the formation of more thermodynamically stable Pb²⁺ complexes (Figure 5). In turn, we hypothesize that the incorporation of the longer OEG₄ linker between the internal and surface layers of compound (3) nanoribbons relative to the OEG₂ linker in compound (2) nanoribbons allows for concomitant spatial distribution of the chelating groups and enhancements in their dynamic behavior, leading to the significant improvement in the Pb²⁺-binding performance of the materials (Figure 5).

Fig. 5 | Hypothesized nanoribbon surfaces illustrating how the addition and lengthening of oligo(ethylene glycol) linkers in the design of amphiphiles underlying self-assembled nanoribbons enhance surface dynamics, flexibility, and spatial organization to mediate surface Pb^{2+} chelation. The characterization reported in this manuscript suggests that the addition of a short OEG₂ linker between compound (1) and (2) nanoribbons enhances surface and interfacial water dynamics to improve Pb^{2+} binding but maintains chelating head groups in close proximity. The extension of this linker to OEG₄ in compound (3) nanoribbons combines enhancements in surface dynamics with additional spatial flexibility to enable each chelating head group to bind Pb^{2+} ions, resulting in a drastic improvement in Pb^{2+} remediation.

Reviewers' comments:

Reviewer #1 (Remarks to the Author):

After careful consideration of the content, I believe that the manuscript should be published as the authors have made significant improvements to address my concerns.

Reviewer #2 (Remarks to the Author):

My comments have been appropriately addressed. I'd like to suggest the publication as is.

Reviewer #3 (Remarks to the Author):

The authors have revised the manuscript considering some, but not all of the comments raised by the referees. I am still not enthusiastic about this manuscript to be published in Nature Communications for the reasons given in my previous report: limited novelty, lack of generality and incremental, specialized work by the same authors. With regards to the scientific revision, I summarize below my comments:

Point 1 from my previous report is fully addressed.

Point 2 has not been addressed and the previous questions remain open. In my opinion, there is still no sufficient evidence of co-assembly. The only experiment that might suggest interaction between the two monomers is EPR, but the signal of the mixture can also originate from homoassemblies of 1-3 that non-specifically/randomly interact with the spin-labelled compounds (the authors mention that these compounds freely dissolve in water). According to this information: remain 4-6 monomeric in water? The co-assembly part remains speculative and requires further elaboration considering my previous suggestions.

Point 3: The authors' revision and the methods used are not related to the comment that was raised, which was the molecular packing of the AA monomers within the fibres. From all suggested measurements, only FTIR was performed. Assuming that the newly conducted FTIR studies (Suppl. Fig 7) are reliable, it appears that there is a high disorder in the hydrogen bonding patterns on the basis of the large number of free and bonded amide groups, which does not match with the ordered fibres imaged by cryoTEM. Please revise the new FTIR experiments. My previous suggestions about analyzing pi-pi interactions, as mentioned by the authors in the text, have not been considered. NMR, UV/Vis or emission studies may be helpful.

In point 4, the authors stress the main claims of the manuscript, which in my opinion are not broadly applicable, thus making this manuscript more suitable to a specialized journal, as mentioned above and in my previous report. The selectivity and sensitivity aspects have not been addressed in the revision.

Point 5 has been fully addressed.

RESPONSE TO REVIEWS

Manuscript ID: NCOMMS-23-05266

Title: “Interfacial dynamics mediate surface binding events on supramolecular nanostructures”

Authors: Ty Christoff-Tempesta, Yukio Cho, Samuel J. Kaser, Linnaea D. Uliassi, Xiaobing Zuo, Shayna L. Hilburg, Lilo D. Pozzo, Julia H. Ortony

Reviewer #1 (Remarks to the Author):

After careful consideration of the content, I believe that the manuscript should be published as the authors have made significant improvements to address my concerns.

We thank Reviewer 1 for their time and effort in reviewing our manuscript.

Reviewer #2 (Remarks to the Author):

My comments have been appropriately addressed. I'd like to suggest the publication as is.

We thank Reviewer 2 for their time and effort in reviewing our manuscript.

Reviewer #3 (Remarks to the Author):

The authors have revised the manuscript considering some, but not all of the comments raised by the referees. I am still not enthusiastic about this manuscript to be published in Nature Communications for the reasons given in my previous report: limited novelty, lack of generality and incremental, specialized work by the same authors. With regards to the scientific revision, I summarize below my comments:

Point 1 from my previous report is fully addressed.

Point 2 has not been addressed and the previous questions remain open. In my opinion, there is still no sufficient evidence of co-assembly. The only experiment that might suggest interaction between the two monomers is EPR, but the signal of the mixture can also originate from homoassemblies of 1-3 that non-specifically/randomly interact with the spin-labelled compounds (the authors mention that these compounds freely dissolve in water). According to this information: remain 4-6 monomeric in water? The co-assembly part remains speculative and requires further elaboration considering my previous suggestions.

We disagree with Reviewer 3's claims that co-assembly is not supported by our data – co-assembly of spin labeled nanostructures is proven explicitly in the manuscript (Page 7) and supplementary materials (especially Supplementary Figure 9). Further, Page 15 of the manuscript (the Methods section) details the extreme care we take in preparing spin labeled nanoribbons to ensure that uniform co-assembly takes place. This method is one of the only direct ways to analyze co-assembly of spin labeled supramolecular structures and has been established for nearly a decade (for example, *Ortony et al. "Internal dynamics of a supramolecular nanofibre." Nature Materials 13.8 (2014): 812*). We understand that interpreting CW EPR spectra of nitroxide spin labels is a relatively uncommon area of expertise; however, we provide guiding information and references to understand the EPR spectra in the manuscript.

Point 3: The authors' revision and the methods used are not related to the comment that was raised, which was the molecular packing of the AA monomers within the fibres. From all suggested measurements, only FTIR was performed. Assuming that the newly conducted FTIR studies (Suppl. Fig 7) are reliable, it appears that there is a high disorder in the hydrogen bonding patterns on the basis of the large number of free and bonded amide groups, which does not match with the ordered fibres imaged by cryoTEM. Please revise the new FTIR experiments. My previous suggestions about analyzing pi-pi interactions, as mentioned by the authors in the text, have not been considered. NMR, UV/Vis or emission studies may be helpful.

We agree with the reviewer that molecular packing is important, and in response, we carried out new and extensive synchrotron X-ray experiments, which are state-of-the-art for determining molecular packing in aqueous assemblies. The revised manuscript provides exhaustive analysis of these synchrotron SAXS, *in situ*, and WAXS profiles to provide quantitative descriptions of molecular organization. The updated manuscript also includes new FTIR spectroscopy, as requested by Reviewer 3, which supports our findings and corroborates the X-ray data. We acknowledge that and apologize for not directly responding to the other suggestions for characterization, namely 2D NMR or UV-Vis spectroscopy. We selected alternative strategies than 2D NMR spectroscopy because this method requires the introduction of a non-water solvent, which we have previously observed to trigger geometric transformations of amphiphilic molecular assemblies in some cases. Similarly, we felt that UV-Vis did not meaningfully complement the performed experimentation because it does not intrinsically provide molecular packing information. The methods that we chose, in combination with nanostructure visualization by cryogenic and conventional transmission electron microscopy, clearly and quantitatively elucidate the molecular packing of AA nanostructures, and therefore we feel that while we did not implement all of the specific techniques suggested by this reviewer, we chose more cutting-edge alternatives that are also more appropriate, given the intricacies of supramolecular materials.

In point 4, the authors stress the main claims of the manuscript, which in my opinion are not broadly applicable, thus making this manuscript more suitable to a specialized journal, as mentioned above and in my previous report.

We feel strongly that our manuscript is broadly applicable, as indicated by the recommendations of Reviewers 1 and 2 to publish as is. The reason why we believe our manuscript has high impact is as follows: Biological systems are well-known to harness conformational and water dynamics as a method for controlling surface chemistry – for example at cell membrane proteins' active sites. In contrast, synthetic materials are also designed to perform chemical reactions, which is an incredibly powerful approach, due to their high surface areas, but the relationship between dynamics and reactivity in synthetic systems is not well-understood or studied with quantitative dynamics information. Therefore, by revealing relationships between dynamics quantified at the nanostructure surface and the potency of surface-tethered moieties (chelators, in our model system), we provide new knowledge for design of highly specialized synthetic nanomaterials. This broad applicability is touched on throughout our manuscript, especially in the last part of the abstract, in the majority of the introduction, and also summarized on page 13: “These results suggest that the conformational dynamics of the molecules that constitute a nanostructure, as well as the dynamics of surface water, can be harnessed to augment chemical events at the interface between a nanomaterial and its aqueous environment.” We feel that we have fully justified the impact of our findings while refraining from exaggerating or using sensational language.

The selectivity and sensitivity aspects have not been addressed in the revision.

The aim of developing a new, highly selective chelating moiety was not a goal of our work and cannot be achieved by typical coordination chemistry; in fact, this goal is a research field in itself. We do feel, however, that those working to develop highly selective chelating moieties could benefit greatly from our conclusions, as we provide a new strategy for improving sensitivity of surface-bound chelators, which reinforces the broad applicability of this manuscript.

Point 5 has been fully addressed.

We thank Reviewer 3 for their time and effort in reviewing our manuscript.

REVIEWER COMMENTS

Reviewer #3 (Remarks to the Author):

I have revised the new version of the manuscript and the authors' reply to my comments. I respectfully disagree with some of the replies, for instance regarding co-assembly (where the type of co-assembly has not been examined) or the use of "more cutting-edge alternatives" to elucidate the molecular packing. None of the used methods gives detailed information about the spatial distribution of atoms and interatomic interactions, as for example MAS or heteronuclear correlation NMR would do. The other aspects concerning the selectivity have also not been addressed. The same applies to the novelty and impact of the manuscript, which is in my opinion incremental work and thus not suitable for Nature Communications.

This being said, it is not my intention to oppose publication of this manuscript. The recommendation of the other two referees is clear and from my side, even if I disagree, I accept the overall recommendation of accepting this paper for publication. I also acknowledge the time and effort invested by the authors in improving the manuscript, even if some of my comments and suggestions still remain open. From my side, there is no need to delay publication of this paper further on the basis of the recommendation of the other two referees.

Reviewer #4 (Remarks to the Author):

In their article "Interfacial dynamics mediate surface binding events on supramolecular nanostructures", Ty Christoff-Tempesta and co-authors report on the influence of the internal flexibility of supramolecular assemblies on the ability of surface chelating groups to bind Pb²⁺ in water. The main focus is on the influence of oligo(ethylene glycol) (OEG) introduced as a linker between the chelating head group and the aramid amphiphiles (AA) responsible for the formation of the nanostructures.

In the following I comment on the use of electron paramagnetic resonance (EPR) for studies in the internal dynamic of the nanostructures. For the investigation of the local dynamic the authors used AAs terminated with paramagnetic TEMPO spin labels instead of the chelating groups. 5 % of the AAs were labelled with different OEGs and self-assembled with their unlabeled counterparts.

Depending on the length of the OEGs, liquid solution EPR spectra change. By simulating the EPR spectra, the authors obtain the rotational correlation rates (DR) and conclude that the spin labels move faster with increasing linker length. This approach is well established for supramolecular assemblies such as biological membranes and the methods used, in particular the simulation of the EPR spectra using the stochastic Liouville equation, is state of the art.

Nevertheless, the analysis of the spectra is not conclusive in some places.

In Figure 9 of the SI, the authors show the EPR spectra of 100% labelled samples and write on page 7 of the main text:

“We note that TEMPO spin-labeled AAs freely dissolved in water display three distinct peaks from isotropically tumbling nitroxide radicals (Supplementary Figure 9).”

I assume it should read “isotropically tumbling” at this point. More importantly, no explanation is given as to why the labels rotate freely. This is unexpected and needs to be investigated further as it could indicate a change in the dynamic of the investigated structures due to the insertion of the labels.

Since the use of the labels assumes that their introduction does not change the overall dynamics, this relationship must be investigated in more detail. Therefore, the authors should show the EPR spectra for all three samples (3-6, 2-5 and 1-4) with a stepwise change in the concentration of 4, 5 and 6 from 1 % to 100 %. In addition, the expected broadening of the EPR spectra, which was already observed by the authors when comparing the 5 and 10 % samples, would provide information about the mean distance of the labels and the co-assembly of labelled and unlabelled AAs.

Finally, I would like to ask the authors to add information on the details of the spectral simulations to the SI, in particular, an evaluation why the ERP spectra can be modelled by a simple scalar rotation rate although the binding to the AAs should render the motion strongly anisotropic.

RESPONSE TO REVIEWS

Manuscript ID: NCOMMS-23-05266

Title: “Interfacial dynamics mediate surface binding events on supramolecular nanostructures”

Authors: Ty Christoff-Tempesta, Yukio Cho, Samuel J. Kaser, Linnaea D. Uliassi, Xiaobing Zuo, Shayna L. Hilburg, Lilo D. Pozzo, Julia H. Ortony

Reviewer #4 (Remarks to the Author):

In their article “Interfacial dynamics mediate surface binding events on supramolecular nanostructures”, Ty Christoff-Tempesta and co-authors report on the influence of the internal flexibility of supramolecular assemblies on the ability of surface chelating groups to bind Pb^{2+} in water. The main focus is on the influence of oligo(ethylene glycol) (OEG) introduced as a linker between the chelating head group and the aramid amphiphiles (AA) responsible for the formation of the nanostructures.

In the following I comment on the use of electron paramagnetic resonance (EPR) for studies in the internal dynamic of the nanostructures. For the investigation of the local dynamic the authors used AAs terminated with paramagnetic TEMPO spin labels instead of the chelating groups. 5 % of the AAs were labelled with different OEGs and self-assembled with their unlabeled counterparts.

Depending on the length of the OEGs, liquid solution EPR spectra change. By simulating the EPR spectra, the authors obtain the rotational correlation rates (DR) and conclude that the spin labels move faster with increasing linker length. This approach is well established for supramolecular assemblies such as biological membranes and the methods used, in particular the simulation of the EPR spectra using the stochastic Liouville equation, is state of the art.

Nevertheless, the analysis of the spectra is not conclusive in some places.

In Figure 9 of the SI, the authors show the EPR spectra of 100% labelled samples and write on page 7 of the main text: “We note that TEMPO spin-labeled AAs freely dissolved in water display three distinct peaks from isotopically tumbling nitroxide radicals (Supplementary Figure 9).” I assume it should read “isotropically tumbling” at this point. More importantly, no explanation is given as to why the labels rotate freely. This is unexpected and needs to be investigated further as it could indicate a change in the dynamic of the investigated structures due to the insertion of the labels.

Thank you for catching this typo! We have updated “isotopically” to “isotropically.”

We appreciate that this point was lacking context before, so we have performed new experiments and updated the Supplementary Figure for clarity. Previously, we were attempting to show that the behavior of the (isolated) spin probe-containing molecule at very dilute concentrations behaves as expected, *i.e.* as dissolved probes. In preparing this response, we discovered that the EPR spectra

for a dilute spin-labeled AA with a different chemical structure than (6) was previously used as the “only (6)” control due to a miscommunication between authors. Consequently, we have verified that all of the data throughout the manuscript and SI is correctly labeled and note that no other instances like this were identified, and we deeply appreciate that this was caught during the review stage. To address the reviewer’s question regarding the behavior of only spin-labeled AAs, we have replaced the previous spectrum with EPR spectra of spin probe-containing AA molecules in either water (which allows for aggregation) or a mixture of acetonitrile and water (in which the molecules are dissolved). These experiments are now performed using spin-labeled AA concentrations at the same concentration used in the co-assembly experiments. We believe these experiments more clearly demonstrate the distinction between the behavior of the spin probes in isolation versus in the assembly, and have copied the corresponding changes to the manuscript below:

[Main text]: We note that TEMPO spin-labeled AAs freely dissolved in a mixture of acetonitrile and water display three distinct peaks from isotropically tumbling nitroxide radicals, while TEMPO spin-labeled AAs suspended in only water display a single, very broad peak arising from spin probe interactions indicative of molecular aggregation (Supplementary Figure 10).

[Supplementary Information]:

Supplementary Figure 10. EPR spectra of (a) a suspension of compound (6) in 50:50 (by vol.) acetonitrile:water; and aqueous suspensions of (b) compound (6), (c) 5 mol% of compound (6) in compound (3), and (d) compound (3). Spectras (a) and (b) contain the same molar concentration of (6) in solvent as the concentration of (6) in spectra (c). The suspension of compound (6) in acetonitrile/water shows a free-tumbling lineshape indicative of fully dissolved molecules, and the aqueous suspension of compound (6) demonstrates significant broadening from head group aggregation. The mixture of compound (6) in (3) exhibits a macroscopic order-microscopic disorder-like lineshape, implying co-assembly. In contrast, the aqueous suspension containing only compound (3) displays no EPR signal.

Since the use of the labels assumes that their introduction does not change the overall dynamics, this relationship must be investigated in more detail. Therefore, the authors should show the EPR spectra for all three samples (3-6, 2-5 and 1-4) with a stepwise change in the concentration of 4, 5 and 6 from 1 % to 100 %. In addition, the expected broadening of the EPR spectra, which was already observed by the authors when comparing the 5 and 10 % samples, would provide information about the mean distance of the labels and the co-assembly of labelled and unlabelled AAs.

In response to this comment, we have added the following concentration series with systematically varied spin label concentrations and its corresponding context to the Supplementary Information:

Supplementary Figure 9. Concentration series varying the mol% of spin-labeled compounds (legend) in co-assemblies with the corresponding unlabeled amphiphile. **(a)** Co-assembly of compound **(1)** with varying mol% of compound **(4)** (indicated in legend). **(b)** Co-assembly of compound **(2)** with varying mol% of compound **(5)** (indicated in legend). **(c)** Co-assembly of compound **(3)** with varying mol% of compound **(6)** (indicated in legend). Co-assemblies with 5 mol% of spin label maintain spectral features observed in 2 mol% spin-labelled co-assemblies but with significantly improved signal-to-noise. Co-assemblies with 10 mol% or more spin labelled compounds show evidence of exchange broadening. Spin labels in a 5 mol% co-assembly would be tethered to AAs whose structural domains are on average ~ 10 nm from one another, based on previously determined internal dimensions of AA assemblies.²³ Artifacts at low (~ 3260 G) and high (~ 3345 G) in the 2 mol% spin-labeled samples arise from Mn(II) contamination from the Critoseal used to seal the EPR tubes. Intensities of all spectra are normalized to the height of the center peak (~ 3310 G) for comparison of broadening.

Finally, I would like to ask the authors to add information on the details of the spectral simulations to the SI, in particular, an evaluation why the ERP spectra can be modelled by a simple scalar rotation rate although the binding to the AAs should render the motion strongly anisotropic.

We appreciate that this point wasn't clearly addressed in the manuscript. The CSCA toolkit used to model EPR spectra in the manuscript includes fitting for spin probe anisotropy, allowing for quantitative comparisons among rotational diffusion constants. We have added the following text to the Methods section and details to the SI to address this concern:

[Methods]: The CSCA toolkit captures potential anisotropy in spin probe motion by fitting for the parallel and perpendicular components of the axial g -tensor alongside D_R and Gaussian (exchange) broadening.^{36,45} Further details are provided in the Supplementary Information.

[Supplementary Information]: EPR spectra, especially of spin labels undergoing slow conformational motion, are prone to overfitting because of the presence of many shallow local minima in the dynamics landscapes.³⁶ To mitigate this, we used the Chi-Squared Cluster Analysis (CSCA) toolkit to modeling the EPR spectra reported in this manuscript. EPR spectra were fit for the parallel and perpendicular components of the axial g -tensor to account for spin label anisotropy, the rotational diffusion constant (D_R), and Gaussian (exchange) broadening. The selection of these fitting variables was chosen to accurately model the experimental spectra while using as few variables as possible to minimize overfitting, allowing for quantitative comparisons of the reported D_{RS} . The CSCA program has been used elsewhere to accurately model macroscopic order-microscopic disorder-type systems where spin label anisotropy would be expected.⁴⁵

REVIEWER COMMENTS

Reviewer #4 (Remarks to the Author):

In their revised manuscript, the authors addressed my questions and added revised Figures 9 and 10 to the supplementary information. Most of the points I had raised in my previous review, have been sorted out.

There are two remaining points, which I still do not fully understand. In Figure 9 of the revised SI the authors are showing a dilution series between 2 % to 50 % TEMPO labelled AAs. In Figure 10 of the SI, they are showing an EPR spectrum of 100 % labelled (6) sample. I was surprised to see that while 50 % labelled samples show still quite well resolved spectra the 100 % samples exhibit a single line indicating strongly increased spin-spin interactions. Was this strong change between the 50 % and 100 % samples observed for all three compositions 1-4, 2-5 and 3-6?

Concerning the description of the spectral simulations, I have a few follow up questions.

On page 14 of the revised SI the authors write:

“EPR spectra were fit for the parallel and perpendicular components of the axial g-tensor to account for spin label anisotropy, the rotational diffusion constant (DR), and Gaussian (exchange) broadening.”

TEMPO has no axial g-tensor and at X-band frequencies, it is both the Hyperfine and the g-tensor, which needs to be considered. In addition, I did not understand how the spectra can be fit for the parallel and perpendicular component of the g-tensor. The authors should clarify this point.

Finally, in my previous review, I had asked the authors to comment on the fact that they use an isotropic correlation time DR, while orientation dependent orientational motion and as a result orientation dependent times are very likely in the present case.

RESPONSE TO REVIEWS

Manuscript ID: NCOMMS-23-05266

Title: “Interfacial dynamics mediate surface binding events on supramolecular nanostructures”

Authors: Ty Christoff-Tempesta, Yukio Cho, Samuel J. Kaser, Linnaea D. Uliassi,
Xiaobing Zuo, Shayna L. Hilburg, Lilo D. Pozzo, Julia H. Ortony

Reviewer #4 (Remarks to the Author):

In their revised manuscript, the authors addressed my questions and added revised Figures 9 and 10 to the supplementary information. Most of the points I had raised in my previous review, have been sorted out.

There are two remaining points, which I still do not fully understand. In Figure 9 of the revised SI the authors are showing a dilution series between 2 % to 50 % TEMPO labelled AAs. In Figure 10 of the SI, they are showing an EPR spectrum of 100 % labelled (6) sample. I was surprised to see that while 50 % labelled samples show still quite well resolved spectra the 100 % samples exhibit a single line indicating strongly increased spin-spin interactions. Was this strong change between the 50 % and 100 % samples observed for all three compositions 1-4, 2-5 and 3-6?

This was an interesting find for us as well. We attribute the change from a well-resolved 3-peak spectrum at 50% (6) in (3) to a single peak at 100% (6) to head group aggregation, or otherwise some sort of disordered partitioning of the hydrophobic and hydrophilic domains of (6) in water. Aggregation in compound (6) most likely results from the differential intermolecular forces across the molecule of (6), however the molecule does not appear to be sufficiently amphiphilic to self-assemble into ordered nanostructures on its own in water. The result of this is an EPR spectrum that reflects strong spin-spin coupling, but no ordered aggregation by structural characterization methods. As for the structure of spin labeled compounds (4) and (5) at 100% in water (with no “filler” non-spin labeled compound), the solubilities of these samples were so low that we were unable to meaningfully analyze these samples by EPR. Considered that (4) and (5) contain fewer ethylene glycol repeats, their low solubility in water on their own is expected. In contrast, compound (6), with the lengthiest OEG linker of the three spin-labeled compounds, was indeed soluble enough to create a suspension for analysis at 100% molar concentration of the spin label.

Concerning the description of the spectral simulations, I have a few follow up questions.

On page 14 of the revised SI the authors write:

“EPR spectra were fit for the parallel and perpendicular components of the axial g-tensor to account for spin label anisotropy, the rotational diffusion constant (DR), and Gaussian (exchange) broadening.”

TEMPO has no axial g-tensor and at X-band frequencies, it is both the Hyperfine and the g-tensor, which needs to be considered. In addition, I did not understand how the spectra

can be fit for the parallel and perpendicular component of the g-tensor. The authors should clarify this point.

Finally, in my previous review, I had asked the authors to comment on the fact that they use an isotropic correlation time D_R , while orientation dependent orientational motion and as a result orientation dependent times are very likely in the present case.

Thank you for catching this – we agree that very little anisotropy is observed in g-tensors for MOMD systems at X-band frequencies. We apologize for our error, which was propagated from the variable descriptions in the fitting toolkit that appear to be erroneously labeled as describing the perpendicular and parallel components of g-tensor anisotropy rather than hyperfine/A-tensor anisotropy. We have updated the manuscript to remove this claim.

Regarding the choice of an isotropic correlation time, we selected a fitting package that minimizes the number of variables used while maintaining a high-quality, descriptive fit because EPR spectra are highly sensitive to overfitting. We appreciate that this wasn't thoroughly discussed before, and have updated the Supplemental Information to include the following:

EPR spectra, especially of spin labels undergoing slow conformational motion, are prone to overfitting because of the presence of many shallow local minima in the dynamics landscapes.³⁶ To mitigate this, we used the Chi-Squared Cluster Analysis (CSCA) toolkit to modeling the EPR spectra reported in this manuscript. EPR spectra were fit for the parallel and perpendicular components of the axial A-tensor to account for spin label anisotropy, the rotational diffusion constant (D_R), and Gaussian (exchange) broadening. The selection of these fitting variables was chosen to accurately model the experimental spectra while using as few variables as possible to minimize overfitting, allowing for quantitative comparisons of the reported D_{RS} . **Fitting X-band EPR spectra of anisotropically rotating systems with isotropic rotational correlation times generally leads to a low-quality fit of the relative amplitudes of h(+1) and h(0) peaks.⁴⁷ However, our spectra were well-modeled by an isotropic correlation time in this respect (Figure 1), so we opted for a simpler motional model to avoid overfitting. An assumption of spherically symmetric diffusion is often consistent with experimental results for nitroxide spin labeled covalently appended into supramolecular/macromolecular ensembles.^{45,46} We note that the CSCA program has been used elsewhere to accurately model macroscopic order-microscopic disorder-type systems where spin label anisotropy would be expected.⁴⁵**

We thank Reviewer 4 for their time and effort in reviewing our manuscript!

REVIEWERS' COMMENTS

Reviewer #4 (Remarks to the Author):

The authors of the manuscript NCOMMS-23-05266D have answered all outstanding questions in a convincing manner. I recommend that their manuscript be published in its present form.